# Investigation on the Usefulness of Sulfamethoxazole Trimethoprim Combination Small Tablets in Pediatric Pharmacotherapy: A Single Center Observational Study Using a Questionnaire

**DOI:** 10.3390/children9101598

**Published:** 2022-10-21

**Authors:** Jumpei Saito, Miho Yamaguchi, Seiichi Shimizu, Kyoko Chiba, Tomoyuki Utano, Akinari Fukuda, Seisuke Sakamoto, Mureo Kasahara, Akimasa Yamatani

**Affiliations:** 1Department of Pharmacy, National Center for Child Health and Development, Tokyo 157-8535, Japan; 2Organ Transplantation Center, National Center for Child Health and Development, Tokyo 157-8535, Japan

**Keywords:** sulfamethoxazole trimethoprim, pediatrics, formulation, small tablet

## Abstract

Sulfamethoxazole trimethoprim (ST) combinations are used to prevent infection in immunocompromised patients. In pediatric patients, conventional ST combination tablets (cTab) are large and granules are not preferred due to their rough and bitter taste in the mouth. Since a new formulation of smaller tablets (sTab, 1 cTab = 1-gram granules = 4 sTab) was approved, a study regarding the usability of sTab in pediatric patients was conducted. Children who started taking sTab of the ST combination at our hospital between August 2021 and August 2022 were included. Using an anonymous questionnaire, the dosage of ST combinations, the child’s response (3-point visual scale: positive, neutral, or negative), preparation and administration time, and method of taking the drug were asked. Twenty-two patients (median age: 11.0 years) receiving cTab. Median (range) number of tablets per dose was 1 (0.5–1.5) tablet, and was 4 tablets (1.0–4.0) after switching to sTab. Twenty patients (median age: 5.0 years) receiving granules. Median (range) single dose was 0.75 (0.2–2.0) gram, and was 2.0 (1.0–4.0) tablets after switching to sTab. Post-dose reactions were positive in 5, neutral in 7, and negative in 10 cases for cTab, and positive in 1, neutral in 7, and negative in 12 cases for granules. After switching to sTab, 9, 13 and 0 cases, and 10, 9 and 1 cases were positive, neutral, and negative, respectively. Median preparation and administration times were decreased after switching to sTab in both cTab and granules groups. The frequency of dosage manipulations was also decreased. The switch to sTab improved acceptability, and decreased burden of administration, suggesting that sTab is a user-friendly formulation in pediatric medications.

## 1. Introduction

In oral drug therapy in children, the taste of the medication and the child’s ability to swallow the medication significantly affect the choice and prescription of the medication [1]. Furthermore, in clinical practice, it is essential to consider the ease with which parents prepare medications and the ease with which the affected child can swallow them in order to ensure patient acceptability, compliance, and successful treatment. When appropriate pediatric dosage forms are not commercially available, dose adjustment requires manipulation of adult dosage forms, such as splitting or crushing tablets, opening capsules, dispersing tablets or capsules into liquids, adjusting ratios to adjust dosage, cutting suppositories, and applying injectable solutions through other routes. In addition, tablets may be crushed when dose-adjustable dosage forms, such as granules, are available but are not easy to take. However, such manipulation inevitably affects the safety of the drug product, and thus its bioavailability and pharmacokinetics [1]. Manufacturers are therefore required to provide information on the effects of such manipulations, conduct pediatric studies with pediatric products, and develop dosage forms and drugs suitable for pediatric use.

Sulfamethoxazole trimethoprim (ST) combination products are used to prevent infection in patients with immunosuppressed conditions [2,3]. The main dosage forms are tablets and granules. The conventional tablet (cTab) is large (approximately 11 mm, Figure 1A), and the granules (Figure 1B) have a bitter taste and rough texture in the mouth, which may make them difficult to take [4]. In addition, it is time-consuming to prepare the tablets by dividing them into small pieces before taking, or mixing them with viscous food to reduce bitterness and roughness. A new ST combination drug, the small ST tablet (sTab, Figure 1C), has now been developed, containing 20 mg of trimethoprim and 100 mg of sulfamethoxazole per tablet [5], which is one quarter of cTab and equivalent to 0.25 g of granules. It is approximately 6.0 mm in diameter and 4.4 mm thick. The weight is approximately 0.13 g. sTab is uncoated tablet and has the same additive composition as the cTab except for sucralose (additives include calcium carmellose, hydroxypropyl cellulose, sucralose, and magnesium stearate). Sucralose is added as a sweetener to suppress bitterness. sTab was developed to improve dosing for children and elderly population, but it is unclear whether it improves handling and dosing for both the drug recipient and the child. In this study, a questionnaire survey was used to evaluate the improvement in handling and dosing when the ST formulation was switched from cTab or granules to sTab.

## 2. Materials and Methods

### 2.1. Setting and Participants

This single-center observational study included children aged 0 to 18 years who were prescribed ST combination cTab or granules at the National Center for Child Health and Development (NCCHD). Consent to and completion of the questionnaire was given by a parent or guardian on behalf of patients aged 0 to 7 years; patients aged 7 years and older completed the questionnaire by themselves or on behalf of their parents or guardians.

### 2.2. Research Procedures

The study was approved by the Ethics Committee of the NCCHD (approval number: 2021-1163, approval date: 11 March 2021) and the questionnaire survey began in August 2021. Questionnaires were distributed to children who were eligible to switch to sTab from cTab or granules (those who were receiving cTab in divided or crushed form, those who were eligible to switch from granules to sTab in whole numbers, and those who wanted to switch from cTab or granules to sTab). Responses were obtained at the time of preparing and taking the dosage form (cTab or granules) before switching to sTab, and at the time of preparing and taking the sTab, respectively. A statement regarding the presence or absence of consent was included in the preamble of the questionnaire, and the response to the questionnaire was considered as consent to participate in the study. No solicitation was made by the health care provider to switch to sTab.

### 2.3. Questionnaire Structure and Evaluation

A questionnaire (Table 1) was completed by the parent or the child himself/herself at the time of each drug dose. The questionnaire consisted of information on the background of the patient (gender, age, body weight), information on the administration (dosage form and dosage of the ST combination drug, frequency of administration, duration of administration), information on the administration phase (patient’s reaction after the dose, time taken to prepare the dose, time taken to administration, method of administration, any devices, drinks, and food needed for the dose). The preparation time for the ST combination drug was defined as the time it took to remove the dispensed drug from the package and bring it to the mouth. The time taken to crush the drug or dissolve it in a solvent was also included in the total. The time taken to take the ST combination drug was defined as the time from when the prepared drug was started to when it was brought to the mouth until it was all finished. If the patient refused to take the medication, the time was recorded at the point when the drug administration was stopped. The acceptability was assessed on a three-point acceptability scale (Positive, Neutral, Negative) (Table 1). No intervention was made regarding the method of administration. No other pharmacotherapy or medical treatment was involved.

### 2.4. Sample Size and Statistical Analysis

The purpose of the statistical analysis was to evaluate the change in patient’s response after switching dosage forms as the primary endpoint and the change in time to prepare and take the medication as secondary endpoints. Response data from participants who met the entry criteria and completed the evaluation were included in the analysis. Under the null hypothesis, the sample size required for 20% to choose other than “negative” (i.e., “neutral” or “positive”) before the formulation switch and 80% to choose other than “negative” after the switch to smaller tablets, with an alpha error of 0.05 and power of 0.8, was estimated to be 14 persons in each group. Comparison of median between the two corresponding groups was done with a Wilcoxon signed rank test. Post-dose responses (positive, neutral, negative) were not quantified, but frequency distributions were calculated, and changes in the percentage of non-negative responses before and after each drug administration were compared by Fisher’s exact probability test.

### 2.5. Data Handling

All data from the collected questionnaires were entered into a secure, password-protected, researcher-only accessible database system hosted at the NCCHD. The survey was anonymous, and no relevant data other than the questionnaire existed.

## 3. Results

### 3.1. Patients’ Background and Dosing Status

In this study, 22 patients in the cTab to sTab switching group and 20 patients in the granule to sTab switching group were included. All patients or their parents/guardians to whom the questionnaires were distributed responded to the survey (response rate was 100%). Of the 42 cases, 40 (95.2%) were answered by the parents/guardians on behalf of their children. The background of the patient is shown in Table 2. The median (range) age of the group that switched from cTab to sTab and from granules to sTab was 11.0 (5.0–17.0) and 5.0 (1.0–13.0) years, respectively. Median (range) body weight was 25.7 (14.5–65.1) kg and 19.2 (8.4–42.2) kg, respectively. Median (range) dose of ST combination was 1.0 (0.5–1.5) tablets for cTab and 0.75 (0.2–2.0) grams for granules. The number of tablets after switching to smaller tablets was 4.0 (1.0–4.0) and 2.0 (1.0–4.0) tablets, respectively. All patients in both groups (cTab or granules) had been treated with ST combination drugs for at least 1 month. The duration of treatment at the time of the survey after switching to the sTab was less than 1 month in 15 (68.2%) and 16 (80.0%) patients, and 1 to 3 months in 7 (31.8%) and 4 (20%) patients in each group, respectively.

### 3.2. Administration Status

Table 3 showed the administration status of the ST combination drugs. At the time of evaluation, the group switched from conventional tablets to small tablets, both conventional and small tablets were fully taken. In the group that switched from granules to small tablets, one case of refusal to take the granules and one case of partial taking of the small tablets were observed.

### 3.3. Patient’s Reactions before and after Switching to sTab

Patients’ responses were positive in 5 cases (22.7%), neutral in 7 cases (31.8%), negative in 10 cases (45.5%) for cTab, and positive in 9 cases (40.9%), neutral in 13 cases (59.1%) l, and 0 (0.0%) were negative after switching to sTab (*p* < 0.0001, negative vs non-negative, Fisher’s test).

In the group that switched from granules to sTab, 1 case (5%) was positive, 7 cases (35.0%) were neutral, and 12 cases (60.0%) were negative in the granules group, and 10 cases (50.0%) were positive, 9 cases (45.0%) were neutral, and 1 case (5.0%) was negative after switching to sTab. (*p* < 0.0004, Negative vs. non-negative, Fisher’s test).

### 3.4. Dosing Methods

Table 4 shows the conditions under which the ST combinations were taken. Sixteen of the 22 patients took all of cTab at once; three took cTab by splitting them, and three took them by crushing into powder form. None of the patients took cTab by mixing them with water or food. Six patients took a sucrose syrup, one patient took ice cream, and one patient took apple juice at the same time as a vehicle other than water.

After switching from cTab to sTab, 20 of 22 patients took all sTab at once; two patients took more than one tablet at a time; and one patient took one tablet at a time. None of the patients mixed sTab with water or food. Two patients (9.1%) took sucrose syrup and one (4.5%) took apple juice at the same time as a vehicle other than water.

In the group receiving granules, 11 of 20 patients (55.0%) took the drug at one time; 9 patients (45.0%) received the drug in multiple doses.

The solvents used for dissolving and suspending the granules were water in 11 cases (55.0%), sucrose syrup in 5 cases (25.0%), and carbonated water in 1 case (5.0%). Three cases (15.0%) were not dissolved and suspended in a vehicle. Water was used as the vehicle at the time of dosing in 17 cases (85.0%), sucrose syrup in 2 cases (10.0%), and ice cream in 1 case (5.0%).

After switching to sTab, 10 of 20 patients (50.0%) took sTab at one time, 9 patients (45.0%) took each tablet in multiple doses, and 1 patient (5.0%) took sTab after crushing into powder form. Nineteen (95.0%) were taken with water only and one (5.0%) with sucrose syrup.

### 3.5. Changes in Preparation Time

The median preparation times were shown in Figure 2. In the group switched from cTab to sTab, the median (range) preparation time was 30 (10–610) seconds for cTab and 10 (5–610) seconds for sTab (*p* < 0.002, Wilcoxon’s signed rank test). In the group switched from granules to sTab, median (range) preparation time was 70 (20–270) seconds for granules and 20 (10–70) seconds for sTab (*p* < 0.0001, Wilcoxon’s signed rank test).

### 3.6. Changes in Administration Time

The median administration times were shown in Figure 3. In cTab to sTab group, median (range) administration time was 25 (5–130) seconds for cTab and 10 (5–70) seconds for sTab (*p* < 0.004, Wilcoxon’s signed rank test). In granules to sTab group, the median (range) administration time was 70 (20–1990) seconds for granules and 10 (5–1210) seconds for sTab (*p* < 0.0001, Wilcoxon’s signed rank test).

## 4. Discussion

In this study, the burden of dose preparation, post-dose response, and dosing method of cTab or granules were evaluated in pediatric patients prescribed ST combinations. The same was also evaluated after switching to the newly developed sTab.

The results of the study showed that the percentage of patients who regarded their reactions as negative after taking cTab and granules decreased significantly in both groups after switching to sTab. Median preparation and administration time also decreased.

Referring to the method of administration, cTab group no longer had to crush or divide tablets when they were administered sTab. The percentage of cases in which the drug was mixed with water or food prior to dosing remained the same in the tablet group but decreased in the granule group. Sucrose syrup, ice cream, and juice were used as vehicles used at the time of administration, but the frequency of use decreased after switching to sTab.

Preparation time (from taking the drug out of the package to taking it to the mouth) and administration time (from placing the drug in the mouth to swallowing it completely) were also reduced for sTab administration. The absence of processes of crushing, dividing, and mixing the tablets with water or taste masking agents were thought to have contributed to the reduction in preparation time for sTab.

sTab is approximately 6 mm in diameter and 4.4 mm thick compared to cTab (approximately 11 mm in diameter and 5.1–5.3 mm thick). The smaller size of the tablets made them easier to swallow, which was thought to have contributed to the improved post-dose response. In addition, the fact that even one-year-old children were able to take sTab in this study suggests that sTab may be an option for dosage forms that can be taken by younger children. In addition, “roughness” and “large volume” were cited as factors contributing to the difficulty of taking granules [4]. One gram of granule formulation is equivalent to four small tablets but has a volume of approximately 0.52 g. The reduction in the volume of the drug to be taken may have contributed to improved ease of administration. In addition, the ST combination drug has a bitter taste in the active pharmaceutical ingredient. Therefore, if the tablet is kept in the mouth for a long time or the granules dissolve, the bitter taste may be perceived. The spreading or roughness of the granules in the oral cavity can also lead to discomfort when taking the drug [6,7]. The change from granules to sTab is thought to have contributed to the decrease in volume and improvement in the feeling of taking the tablets, which in turn led to improved post-dose reactions. Regarding the size of sTab, sTab is not too small compared to tablets conventionally used in pediatric patients, and no handling problems is expected.

In addition, the artificial sweetener sucralose is added as an additive to sTab. This is thought to contribute to the reduction of bitterness even when the tablets are retained in the oral cavity. Even if sTab themselves cannot be swallowed, the bitterness can be reduced after sTab are crushed.

Disadvantages of using sTab include an increase in the number of tablets, higher drug costs, and reduced dose adjustment like granules. Patients also commented that the tablets were too sweet. Additionally, in Japan, tablets are automatically packaged in a single package by an automatic tablet dispensing machine. Currently, sTab is not available in PTP sheets, but only in bottles, which may complicate counting and dispensing in the pharmacy department. In light of these disadvantages, we believe that consideration should be given to introducing sTab into hospital or pharmacy formularies or changing from granules or cTab to sTab.

Of course, small tablets are not necessarily the optimal solution as a drug for preventing infections in immunocompromised children. Liquid form is another candidate, but they must be dispensed by a parent or nurse and carry the risk of measurement errors [8]. The process of measuring is also complicated for patients who are taking multiple medications. In addition, additives are added to mask bitterness and improve shelf life, but there are also concerns about safety for children [9]. Among tablets, an orally disintegrating dosage form is usually better. However, masking the bitterness of the main drug would require coating of the disintegrated granules, which is unlikely to be willingly developed by pharmaceutical companies due to development costs and current drug prices. Film formulations may also be acceptable but may be difficult to contain for large-volume ST combination ingredients [10,11]. Another dosage form option is sprinkle capsules [12]. This formulation would encapsulate the mini-tablet or granule formulation. While dispensing at the pharmacy is simplified, disadvantages such as increased preparation time for parents and nurses and the need for masking of the contents also exist. Naturally, it should also be noted that development costs would be higher.

The development of new dosage forms to replace granules and large tablets is also required for other pediatric formulations. Although there have been remarkable technological breakthroughs in formulation development, many older drugs, especially those with dosing problems, have expired patents, and development is difficult. To enable patient-centered drug development, it is desirable to expand development support and establish evidence on the usability and acceptability of new dosage forms.

## 5. Conclusions

In pediatric patients, switching to sTab of ST combination drugs improved both acceptability, preparation, and administration time. This may lead to improved adherence.

## Figures and Tables

**Figure 1 children-09-01598-f001:**
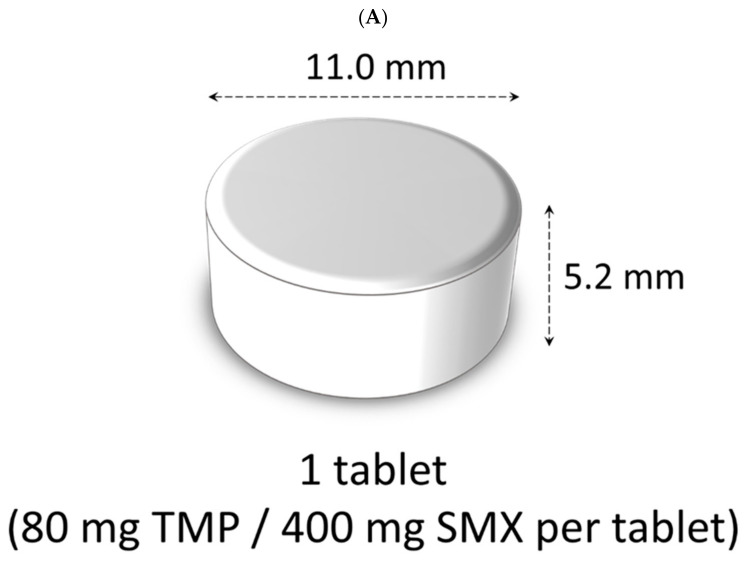
Sulfamethoxazole-trimethoprim combinations. Conventional tablet (**A**) contains 80 mg of trimethoprim (TMP) and 400 mg of sulfamethoxazole (SMX) per 1 tablet. Granules (**B**) contains 80 mg of TMP and 400 mg of SMX per 1 g. Small tablet contains 20 mg of TMP and 100 mg of SMX per 1 tablet. Small tablets (**C**).

**Figure 2 children-09-01598-f002:**
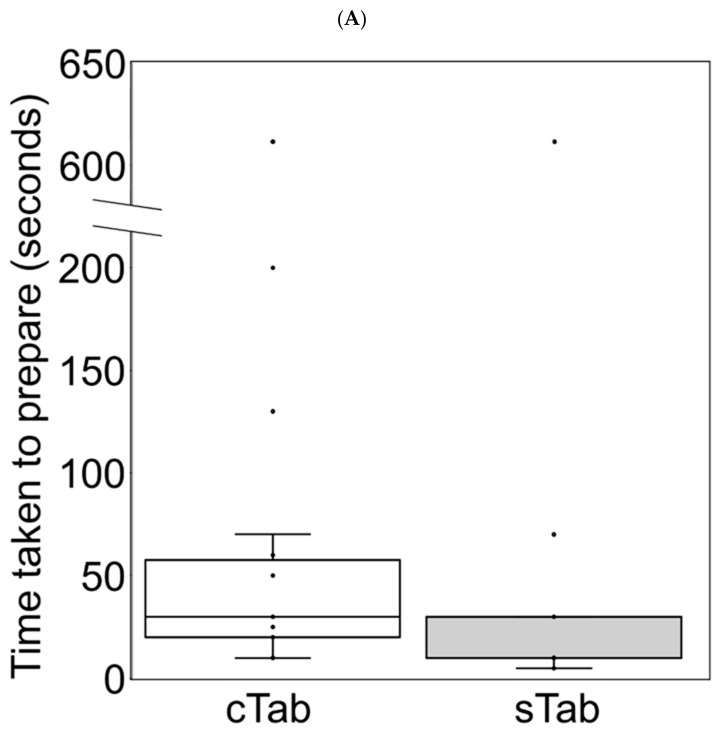
Time taken to prepare the Sulfamethoxazole trimethoprim combinations. Each plot showed the preparation time for each case. The center line indicated the median, boxes indicated first and third quartile range (IQR), whiskers indicated upper or lower fence, and plots outside the fence indicated outliers (1.5 × IQR). White boxes indicate preparation times for conventional tablets (**A**) or granules (**B**); black boxes indicated preparation times for small tablets (**A**,**B**). cTab, conventional tablet; sTab, small tablet.

**Figure 3 children-09-01598-f003:**
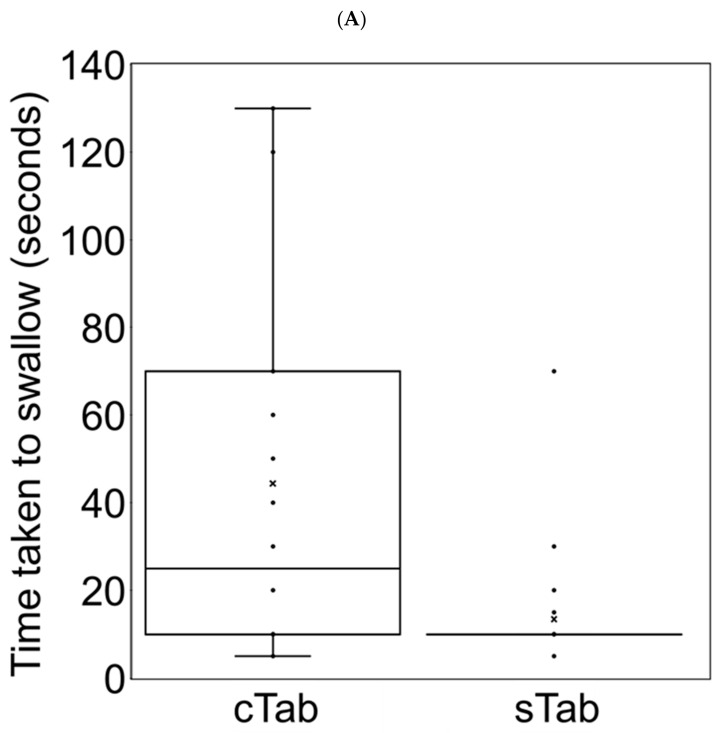
Time taken to administration the Sulfamethoxazole trimethoprim combinations. Each plot showed the administration time for each case. The center line indicated the median, boxes indicated first and third quartile range (IQR), whiskers indicated upper or lower fence, and plots outside the fence indicated outliers (1.5 × IQR). White boxes indicate administration times for conventional tablets (**A**) or granules (**B**); black boxes indicated administration times for small tablets (**A**,**B**). sTab, small tablet.

**Table 1 children-09-01598-t001:** Questionnaire sheet (translated into English).

1. I agree to participate in this survey		
□ Yes		
□ No		
2. The observer in charge of the reports:		
□ A caregiver		
□ A healthcare professional		
□ A patient		
3. The patients		
3-1. What is the gender of the patient?		
□ Girl		
□ Boy		
3-2. What is the age of the patient?		
3-3. What is the patient’s weight?		
4. Dosages for ST combination drugs		
4-1. Which ST combination drug is the patient taking?		
□ Conventional ST combination tablet		
□ ST combination granules		
□ Small ST combination tablets		
4-2. What is the dosage for one dose of ST combination drug?		
4-3. How often should this medication be taken?		
4-4. What is the duration of administration?		
□ For less than 1 month		
□ For 1 month or more		
□ For 3 months or more		
5. The context of use		
5-1. Where was the medication taken?		
□ At home		
□ In the hospital		
□ Other		
5-2. At what time of day was the medication taken?		
6. Observations		
6-1. What was the patient’s reaction when taking the medication?		
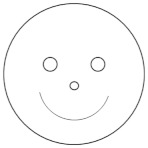	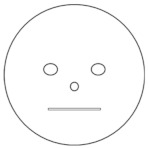	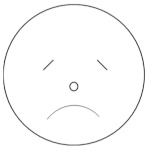
6-2. How much of the medication had been taken?		
□ All of the prescribed dose has been taken		
□ A part of the prescribed dose has been taken		
□ The prescribed dose hasn’t been taken		
6-3. How long did it take to prepare the prescribed dose of medication?		
(starting from the opening of the medication package)		
6-4. What was the administration time of the prescribed dose of medication?		
(starting from the moment it is ready to use)		
6-5. Before dosing, the prescribed dose of the medication had to be...		
□ … modified before administration		
(e.g., prescribed dose of tablet divided or crushed into powder)		
□ … mixed with drink or food		
6-6. When the patient puts it in the mouth, the patient had to…		
□ … take a drink or food to mask the taste or for easier swallowing		
□ … be promised a reward		
□ Other		

**Table 2 children-09-01598-t002:** Patients’ demographics.

	(A) cTab to sTab	(B) Granules to sTab
Subject number (boy, %)	22 (11, 50.0%)	20 (11, 55.0%)
Median age (range)	11.0 (5.0–17.0)	5.0 (1.0–13.0)
Median body weight (kg, range)	25.7 (14.5–65.1)	19.2 (8.4–42.2)
Duration of conventional ST combination treatment		
Less than 1 month	0 (0.0%)	0 (0.0%)
1 to 3 months	2 (9.1%)	5 (25.0%)
More than 3 months	20 (90.9%)	15 (75.0%)
Duration of sTab treatment		
Less than 1 month	15 (68.2%)	16 (80.0%)
1 to 3 months	7 (31.8%)	4 (20.0%)
More than 3 months	0 (0.0%)	0 (0.0%)
Median amount of cTab or granules per dose (range)	1.0 (0.5–1.5) tablets	0.75 (0.2–2.0) gram
Median amount of sTab per dose (range)	4.0 (1.0–4.0) tablets	2.0 (1.0–4.0) tablets

cTab, conventional tablet; sTab, small tablet; ST, Sulfamethoxazole trimethoprim.

**Table 3 children-09-01598-t003:** Patient reactions at the time of dose.

	cTab to sTab		Granules to sTab	
	cTab (n = 22)	sTab (n = 22)	Granules (n = 20)	sTab (n = 20)
(A) Dosing location				
Home	8 (36.4%)	8 (36.4%)	8 (40.0%)	8 (40.0%)
Hospital	14 (63.6%)	14 (63.6%)	16 (60.0%)	16 (60.0%)
(B) Dosing timing				
Morning	20 (90.9%)	20 (90.9%)	19 (95.0%)	19 (95.0%)
Evening	2 (9.1%)	2 (9.1%)	1 (5.0%)	1 (5.0%)
(C) Dosing status				
Fully taken	22 (100.0%)	22 (100.0%)	19 (95.0%)	19 (95.0%)
Partly taken	0 (0.0%)	0 (0.0%)	0 (0.0%)	1 (5.0%)
Not taken	0 (0.0%)	0 (0.0%)	1 (5.0%)	0 (0.0%)
(D) Patient reaction				
Positive	5 (22.7%)	9 (40.9%)	1 (5.0%)	10 (50.0%)
Neutral	7 (31.8%)	13 (59.1%)	7 (35.0%)	9 (45.0%)
Negative	10 (45.5%)	0 (0.0%)	12 (60.0%)	1 (5.0%)

cTab, conventional tablet; sTab, small tablet.

**Table 4 children-09-01598-t004:** Dosing methods and devices.

	cTab to sTab		Granules to sTab	
	cTab (n = 22)	sTab (n = 22)	Granules (n = 20)	sTab (n = 20)
(A) Dosing split and manipulation for intake				
No (taken all at one time)	16 (72.7%)	20 (90.9%)	11 (55.0%)	10 (50.0%)
Divided into two or more	0 (0.0%)	2 (9.1%)	9 (45.0%)	9 (45.0%)
Split into two or more	3 (13.6%)	0 (0.0%)	0 (0.0%)	0 (0.0%)
Crushed into powder	3 (13.6%)	0 (0.0%)	0 (0.0%)	1 (5.0%)
(B) Mixed with food/drink/other masking agents				
No	22 (100.0%)	22 (100.0%)	3 (15.0%)	19 (95.0%)
Dissolved and suspended in water	0 (0.0%)	0 (0.0%)	11 (55.0%)	1(5.0%)
Dissolved and suspended in sucrose syrup	0 (0.0%)	0 (0.0%)	5 (25.0%)	0 (0.0%)
Dissolved and suspended in carbonated water	0 (0.0%)	0 (0.0%)	1 (5%)	0 (0.0%)
(C) Taken with food/drink/other masking agents				
No additional concomitant intake or taken with water only	14 (63.6%)	19 (86.3%)	17 (85.0%)	19 (95.0%)
Taken with sucrose syrup	6 (27.3%)	2 (9.1%)	2 (10.0%)	1 (5.0%)
Taken with ice cream	1 (4.5%)	0 (0.0%)	1 (5.0%)	0 (0.0%)
Taken with apple juice	1 (4.5%)	1 (4.5%)	0 (0.0%)	0 (0.0%)

cTab, conventional tablet; sTab, small tablet.

## Data Availability

Data available on request due to restrictions eg privacy or ethical.

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
