# Peer review of "Investigation on the Usefulness of Sulfamethoxazole Trimethoprim Combination Small Tablets in Pediatric Pharmacotherapy: A Single Center Observational Study Using a Questionnaire"

_children, 2022, doi:10.3390/children9101598_

Round 1

Reviewer 1 Report

The article Ë®Investigation on the Usefulness of Sulfamethoxazole Trimethoprim Combination Small Tablets in Pediatric Pharmacotherapy: A Single Center Observational Study Using a QuestionnaireË®  it is a well-written and well-organized manuscript. It can be accepted in present form.

Author Response

Thank you for spending your valuable time to peer review our paper. We will proofread the English text as appropriate.

Reviewer 2 Report

The article entitled “Investigation on the Usefulness of Sulfamethoxazole Trimethoprim Combination Small Tablets in Pediatric Pharmacotherapy: A Single Center Observational Study Using a Questionnaire “is a well-researched and well written, but I do have some minor issues that can be considered for further improvement: 

Comments

1.Information about new “sTab”is rather limited; For instance, Is it an Orally Disintegrating Tablet’’ or to be prepared in a suitable vehicle

Please provide more details about ‘’sTab’’in the introduction section

2.Handling such small tablet is difficult or not, particularly among elderly and pediatric patient population

Author Response

To Reviewer 2

Thank you for your guidance. We have made the following corrections

Comments to Author:

1.Information about new “sTab”is rather limited; For instance, Is it an Orally Disintegrating Tablet’’ or to be prepared in a suitable vehicle

Please provide more details about ‘’sTab’’in the introduction section

“It is approximately 6.0 mm in diameter and 4.4 mm thick. The weight is approximately 0.13 g.  sTab is uncoated tablet and has the same additive composition as the cTab ex-cept for sucralose (additives include calcium carmellose, hydroxypropyl cellulose, su-cralose, and magnesium stearate). Sucralose is added as a sweetener to suppress bit-terness. sTab is considered bioequivalent to cTab based on dissolution studies.”

2.Handling such small tablet is difficult or not, particularly among elderly and pediatric patient population

“Additionally, in Japan, tablets are automatically packaged in a single package by an automatic tablet dispensing machine. Currently, sTab is not available in PTP sheets, but only in bottles, which may complicate counting and dispensing in the pharmacy department. Considering these disadvantages, we believe that consideration should be given to introducing sTab into hospital or pharmacy formularies or changing from granules or cTab to sTab. Considering these disadvantages, it is necessary to change to smaller tablets.”

Reviewer 3 Report

Overall, the manuscript is scientifically sound and very well-written. However, it could benefit from some improvements on the discussion.

Drug administration to children is a very important topic, and child-friendly formulations is something that we should be working towards, so the work that the authors are doing is much appreciated in the field. However, I think there is much more to discuss about it, which is lacking in this manuscript. For instance, and just to give an example, I would like to read if alternative formulations should be something we should be looking at (see PMID:30071163, technology applies even if the therapy does not directly match). The article will benefit from an improved discussion, tackling important matters in the field.

Author Response

To reviewers,

We deeply appreciate the encouragement of the reviewers. Although it may be difficult to introduce new innovations, especially in the case of ST combination drugs, we have made additions to the "DISCUSSION" section with reference to current innovations.

"Of course, small tablets are not necessarily the optimal solution as a drug for preventing infections in immunocompromised children. Liquid form is another candidate, but they must be dispensed by a parent or nurse and carry the risk of measurement errors [8]. The process of measuring is also complicated for patients who are taking multiple medications. In addition, additives are added to mask bitterness and improve shelf life, but there are also concerns about safety for children [9]. Among tablets, an orally disintegrating dosage form is usually better. However, masking the bitterness of the main drug would require coating of the disintegrated granules, which is unlikely to be willingly developed by pharmaceutical companies due to development costs and current drug prices. Film formulations may also be acceptable but may be difficult to contain for large-volume ST combination ingredients [10, 11]. Another dosage form option is sprinkle capsules [12]. This formulation would encapsulate the mini-tablet or granule formulation. While dispensing at the pharmacy is simplified, disadvantages such as increased preparation time for parents and nurses and the need for masking of the contents also exist. Naturally, it should also be noted that development costs would be higher.

The development of new dosage forms to replace granules and large tablets is also required for other pediatric formulations. Although there have been remarkable technological breakthroughs in formulation development, many older drugs, especially those with dosing problems, have expired patents, and development is difficult. To enable patient-centered drug development, it is desirable to expand development support and establish evidence on the usability and acceptability of new dosage forms."